# Vaginal Microbiota Molecular Profiling in Women with Bacterial Vaginosis: A Novel Diagnostic Tool

**DOI:** 10.3390/ijms242115880

**Published:** 2023-11-01

**Authors:** Alevtina M. Savicheva, Anna A. Krysanova, Olga V. Budilovskaya, Elena V. Spasibova, Tatiana A. Khusnutdinova, Kira V. Shalepo, Natalia R. Beliaeva, Galina Kh. Safarian, Kirill V. Sapozhnikov, Natalya I. Tapilskaya, Igor Yu. Kogan

**Affiliations:** D.O. Ott Research Institute of Obstetrics, Gynecology and Reproductive Medicine, 199034 St. Petersburg, Russia; savitcheva@mail.ru (A.M.S.); krusanova.anna@mail.ru (A.A.K.); o.budilovskaya@gmail.com (O.V.B.); elena.graciosae@gmail.com (E.V.S.); husnutdinovat@yandex.ru (T.A.K.); 2474151@mail.ru (K.V.S.); natascha778@yandex.ru (N.R.B.); marinheira@rambler.ru (K.V.S.); tapnatalia@yandex.ru (N.I.T.); ikogan@mail.ru (I.Y.K.)

**Keywords:** bacterial vaginosis, real-time PCR, *Lactobacillus iners*, *Lactobacillus crispatus*, vaginal microbiota

## Abstract

Bacterial vaginosis (BV) is a most common microbiological syndrome. Multiplex next-generation sequencing (NGS) or molecular tests allow a complete and accurate vaginal microbiota profiling in order to determine the primary causative agent. Due to the high costs and limited availability of NGS, the multiplex real-time PCR draws more attention. The present study aimed to evaluate the microbial composition and dominant lactobacilli species in non-pregnant women with bacterial vaginosis using a multiplex RT-PCR test and determine its diagnostic significance. In total, 331 women complaining of vaginal discharge were included. BV was confirmed upon clinical examination and Nugent criteria. A real-time PCR test was carried out with a new Femoflor test, which identifies opportunistic bacteria, STD pathogens, and some viruses. According to the results, the rate of lactobacilli is significantly reduced in BV-affected patients when compared to healthy women. Moreover, the rate of *L. crispatus* significantly decreases, while the rate of *L. iners* remains high. Among obligate anaerobic bacteria, *Gardnerella vaginalis* was the most prevalent in women with BV. The Femoflor test demonstrated high sensitivity and specificity for diagnosing BV. Moreover, the test allows the identification of infection in women with intermediate vaginal microbiota, as well as STD pathogens, and viruses. Thus, the application of real-time PCR tests can be effectively used in vaginal microbiota evaluation in women with BV, intermediate vaginal microbiota, and healthy women. In addition, this test may be used as an alternative to the Amsel criteria and Nugent scoring method in diagnosing BV.

## 1. Introduction

Bacterial vaginosis (BV) represents a polymicrobial non-inflammatory dysbiosis among reproductively aged women, results in increased susceptibility to viruses and sexually transmitted diseases (STDs), and is associated with a number of gynecological disorders and obstetric complications [1,2,3,4,5,6,7]. The prevalence of BV in women complaining of vaginal discharge reaches 40–50% and ranges from 5.8% to 19.3% in pregnant women [8,9].

Dysbiotic alteration of vaginal microbiota is one of the characteristic signs of bacterial vaginosis. An increased *Gardnerella* spp. number is associated with a biofilm development on the surface of vaginal epithelium, with consequent addition of other anaerobic microorganisms associated with BV via synergistic interactions (*Fannyhessea vaginae* (formerly known as *Atopobium vaginae*), *Prevotella bivia*, *Mobiluncus* spp., *Peptostreptococcus anaerobius*, *Megasphaera* spp., and others) [10,11,12,13,14].

Based on gene sequencing analysis, vaginal microbial communities were divided into separate categories determined by their composition, so-called community state type (CST) [15]. *Lactobacillus crispatus* (CST I) dominance is considered an optimal condition due to its association with reduced susceptibility to STDs [16]. CST III is dominated by *Lactobacillus iners* and is often considered a transitional phenotype to CST IV, and presented with mixed anaerobic microorganisms similar to those found in bacterial vaginosis and is related to an increased risk of reproductive disorders and STD susceptibility. Microbial communities dominated by *Lactobacillus gasseri* (CST II) and *Lactobacillus jensenii* (CST V) are less common than CST I and CST III and are considered opportune vaginal microbiota [15,16].

Both clinical and laboratory methods are used in the diagnosis of bacterial vaginosis in pregnant and non-pregnant women. The most well-known assessment tool is based on the determination of the Amsel clinical criteria (discharge with an unpleasant odor, a positive test with 10% KOH, pH > 4.5, and the presence of “clue” cells at wet mount microscopy). The microscopic method with Gram staining of smears is more specific for the diagnosis of BV than Amsel’s criteria [17]. The most commonly used standardized system for diagnosing bacterial vaginosis is the Nugent scoring method, based on a standardized scoring system that uses the most reliable morphotypes from the vaginal smear [18]. Although, worldwide, both methods have been considered the “gold standard” for the diagnosis of bacterial vaginosis for almost three decades, there are some limitations. For instance, the obtained results often vary since the assessment of diagnostic criteria depends on the skills and expertise of the researcher. Therefore, nowadays, the application of accurate, easy-to-use tests represents a critical approach to the diagnosis of bacterial vaginosis, especially in resource-limited settings.

From this perspective, the introduction of molecular methods for the diagnosis of BV have a number of advantages over the described ones, since they are reliable, determine the number of bacteria, and are optimal for the self-collection of vaginal samples. These technologies offer higher performance and are based on the detection and amplification of specific bacterial nucleic acids. Multiplex nucleic acid amplification tests, such as real-time polymerase chain reaction (RT-PCR), are currently used to diagnose BV by identifying a large number of microorganisms. The ability to amplify more than one target sequence at the same time is one of the main advantages, which is important given the polymicrobial nature of BV. In addition, multiplex PCR allows the evaluation of the quantitative composition and microorganism ratio in a sample [19,20].

Thus, the present study aimed to evaluate the microbial composition and dominant lactobacilli species among non-pregnant women with bacterial vaginosis using a molecular multiplex analysis and determine its diagnostic significance.

## 2. Results

The application of a multiplex molecular test revealed a wide range of microorganisms. Table 1 contains data on microorganisms’ detection rate in the vaginal biotope of women with different types of microbiota according to Nugent’s criteria. 

Lactobacilli are the most important derivatives for maintaining vaginal health. As a rule, two or more vaginal species are present in the same woman, with *L. crispatus* and *L. iners* being most frequently detected. Lactobacilli were verified in all controls and in 99% of women with normal vaginal microbiota (Nugent 0–3). *L. crispatus* was more common in healthy women and women with normal vaginal microbiota, while *L. iners* were more abundant in women with normal vaginal microbiota (64.5%) and in women with BV (56.1%). Bifidobacteria were detected three times more often in women with BV. In terms of facultative anaerobic microorganism presence, the most significant differences were observed in the detection rate of *Haemophilus* spp. in women with BV in comparison to controls. Anaerobes are the predominant microorganisms in women with BV, with particularly high occurrence of *Gardnerella vaginalis*, *Atopobium vaginae,* and *Megasphaera* spp./*Veilonella* spp./*Dialister* spp. when compared to healthy women. *Candida albicans* was more often found in women with BV, while *Candida non-albicans*, on the contrary, was detected only in patients with normal vaginal microbiota. Commensal pathogens such as *Ureaplasma parvum* were determined in all groups; *Ureaplasma urealyticum*—only in the main group; and *Mycoplasma hominis*—exclusively in the group of patients with BV. *Mycoplasma genitalium*, *Chlamydia trachomatis*, and *Neisseria gonorrhoeae* were revealed only in patients with BV. HPV was found in each group, Herpes simplex virus was detected in women with normal vaginal microbiota and intermediate scoring, while Cytomegalovirus was present in the control group, as well as in women with normal vaginal microbiota and intermediate scoring.

We carried out a quantitative assessment of detected species in different types of vaginal microbiota in accordance with Nugent’s criteria by assessing the total bacterial count and the obligate anaerobes, facultative anaerobes (aerobes), and lactobacilli. The bacterial load is shown in Figure 1.

The microbial community, represented by obligate anaerobic microorganisms, is predominant in BV. However, under normal conditions, anaerobic microorganisms are also detected. We evaluated the detection rate and the number of obligate anaerobes in women with different types of vaginal microbiota (Figure 2).

Furthermore, we conducted a comparative analysis between lactobacilli-positive and lactobacilli-negative samples in patients from the control and main groups in order to assess the effect of the lactobacilli species on vaginal microbiota and women’s health. For this purpose, all samples were divided into 4 groups: control, negative, intermediate, and BV. In samples where 2 or more species of lactobacilli were present, we identified the dominant members. As a result, all control group samples refer to normal vaginal microbiota but differ from the “negative” subgroup by the absence of complaints (Figure 3).

Vaginal lactobacilli species detected in women with BV were of particular interest. Thus, we conducted a comparative analysis of the dominant members by dividing all samples into two groups according to the presence or absence of BV (Figure 4).

In order to estimate the effect of the dominant microbiota in the prevention of opportunistic infections, STDs, and viruses, we assessed the detection rate of these microorganisms in patients with and without BV. Figure 5 contains data on the prevalence of *Candida* spp., *Candida albicans*, *Ureaplasma urealyticum*, *Ureaplasma parvum*, and *Mycoplasma hominis* according to the presence or absence of BV.

STD pathogens (*Mycoplasma genitalium*, *Chlamydia trachomatis*, *Neisseria gonorrhoeae*, and *Trichomonas vaginalis*) were identified only in women with BV. Viral agents (HSV1, HSV2, CMV, and HPV) were detected in a few cases in all groups.

It should be noted that, in total, the Femoflor test determined 14 high-risk HPV subtypes: 16, 18, 31, 33, 35, 39, 45, 51, 52, 56, 58, 59, 66, and 68. The incidence of HPV in women complaining of discharge varied between 13.3−26.5% and was comparable in patients with different types of vaginal microbiota (BV and intermediate type—21.2%, respectively, normal microbiota—26.5%). This parameter was slightly lower in healthy women—13.3%.

Additionally, we evaluated the accuracy of the new Femoflor test in BV prediction by applying a receiver operating curve (ROC) analysis. The predictive approaches are the following: if the decimal logarithm of the lactobacilli/total bacterial count ratio is less than 20% and the decimal logarithm of the anaerobes/total bacterial count ratio is more than 90%, BV is confirmed. If at least one criterion is not met, the BV is not proven. The results of the ROC analysis are presented in Figure 6 and Table 2. The results obtained indicate the high diagnostic significance of the Femoflor test.

According to the quantitative PCR, 56 of 66 BV samples previously confirmed by the Nugent’s criteria (sensitivity—84.8%) were identified. In addition, the absence of BV was correctly determined in 253 of 263 samples with negative microscopy for BV (specificity—96.2%) (Table 3).

Thus, the application of the real-time PCR test allows the identification of the vaginal microbiota in women with BV, without BV, and in healthy women. The molecular profiling of the vaginal microbiota suggests a reliably lower lactobacilli detection rate in BV-positive women when compared to BV-negatives. Markedly, the detection rate of *L. crispatus* is significantly reduced, while the rate of *L. iners* remains quite high. Furthermore, the composition of lactobacilli, bifidobacteria, as well as facultative anaerobic microorganisms in the vaginal biotope does not differ between BV-positive and healthy women. However, a completely different situation is observed in the case of obligate anaerobic microorganisms’ presence. For instance, *Gardnerella vaginalis*, which is found in 99.5% of women affected by BV, has absolute superiority. The same trend is observed in other anaerobic members (*Mobiluncus* spp., *Atopobium vaginae* (*Fannyhessea vaginae*), *Anaerococcus* spp., *Bacteroides* spp./*Porphyromonas* spp./*Prevotella* spp.; *Sneathia* spp./*Leptotrihia* spp./*Fusobacterium* spp.; *Megasphaera* spp./*Veilonella* spp./*Dialister* spp.; *Clostridium* spp./*Lachnobacterium* spp.; *Peptostreptococcus* spp.).

Additionally, due to its high sensitivity and specificity, the Femoflor test may serve as a useful alternative tool to the Amsel and Nugent methods for BV diagnosis. Confirmation of vaginal infection in women with intermediate vaginal microbiota as well as STD pathogens and virus identification represent additional advantages of the test. 

## 3. Discussion

According to the molecular analysis performed in the present study, a high diversity of bacterial species and high bacterial load are present in BV-affected women. With high sensitivity and specificity, BV was confirmed based on a combination of a decreased number of lactobacilli and an increased number of anaerobic bacteria or groups of bacteria. The list of species within the molecular test allowed for the identification of different types of vaginal microbiota. Thus, samples obtained from women without BV were characterized by a relatively homogeneous bacterial composition, mainly *Lactobacillus* spp.

Until recently, the diagnosis of BV was almost entirely based on relatively simple diagnostic methods: a combination of clinical symptoms and laboratory tests. 40 years ago, the Amsel criteria were developed (pathological gray discharge, vaginal pH > 4.5, positive amine test, and the presence of “clue” cells). If three out of four criteria are met, the diagnosis of BV is established [21]. A few years later, a Nugent scoring method for assessing vaginal microbiota was introduced [18], turning it into the gold standard for BV diagnosis. Nowadays, despite obvious limitations, the Nugent method and Amsel criteria remain generally accepted standards for BV evaluation [19].

At present, the introduction of modern diagnostic methods for BV detection among reproductively aged women is of high relevance. Bacterial vaginosis is a widespread condition significantly affecting the quality of life with a high recurrence rate. A hallmark feature of BV is the presence of a polymicrobial biofilm on the vaginal epithelial surface. Fluorescence in situ hybridization (FISH) demonstrated that polymicrobial vaginal biofilm, dominated by *Gardnerella* spp., disrupts epithelial homeostasis and promotes co-infection. Standard antibiotic therapy has low efficacy against biofilm, leading to a high recurrence rate (more than 50% within 12 months after the treatment) [22]. Multiplex next-generation sequencing (NGS) allows a complete and accurate vaginal microbiota profiling, significantly improving the clinical aspect of BV [23]. However, due to its high cost has limited application in routine laboratory practice. The introduction of a PCR-based diagnostic test can also serve as an accurate diagnostic tool for the effective treatment of BV and reproductive health preservation [24].

The diversity of microorganisms identified via new multiplex molecular tests allows the assessment of vaginal microbiota in reproductively aged women complaining of discharge. Along with the BV confirmation, STD pathogens were detected, which in turn significantly influenced the treatment approach. In our study, STD pathogens were detected solely in women with BV, which is consistent with other results, indicating vaginal microbiota vulnerability in the presence of BV-associated bacteria [25].

In our study, *Candida* spp. were more often detected in women with BV, which is possibly explained by the low number or complete absence of lactobacilli in these patients. The latter are known to protect against fungal infection due to their anti-*Candida* activity [26]. In the present study, *Mycoplasma hominis* was found only in women with BV, *Ureaplasma urealyticum* was equally present in all patients with complaints and was absent in healthy women, and *Ureaplasma parvum* was detected in all participants including healthy women. However, it should be noted that the incidence of *Ureaplasma urealyticum* and *Ureaplasma parvum* was higher in women with BV. Our results are in line with a previous Australian study, indicating exclusively *M. hominis* association with BV symptoms in non-pregnant women [27].

The incidence of HPV infection in our study turned out to be lower in women with BV when compared to other participants. Among women with BV, no cases of HSV infection were noted. Our result is opposite to the common conclusion on increased risk of viral infection in the setting of disrupted vaginal microbiota [28]. However, our result can be possibly explained by the small sample size.

Our test allowed the determination of the total number of Lactobacilli, along with the four main vaginal Lactobacilli species. *L. iners* and *L. crispatus* were the most common species detected, which is in concordance with the generally accepted classification for vaginal microbial communities based on the composition of the dominant lactobacilli species [29]. Our results are in line with other studies, confirming *L. crispatus* as a reliable marker of the physiological vaginal microbiota found in healthy women [30,31,32]. At the same time, *L. iners* is commonly found in different vaginal microbiota and is often associated with dysbiosis, including asymptomatic vaginal infections [33,34]. It is worth noting that *L. gasseri* and *L. jensenii* were less frequently detected. The incidence of *L. gasseri* was comparable between different groups. The microbial communities predominated by *L. gasseri* (CST II) are more dynamic in contrast to CST I (*L. crispatus*) and CST V (*L. jensenii*). *L. gasseri* predominance is associated with a vaginal pH of 4.4, which is slightly higher than that in the communities predominated by *L. crispatus* (pH = 4.0) and *L. jensenii* (pH = 4.2). However, such a microbial community is not associated with dysbiosis, in contrast to CST III (*L. iners*, pH > 4.5) [35]. *L. jensenii* was reliably less frequently detected in BV-positive women when compared to healthy women. Both *L. jensenii* and *L. crispatus* produce D−lactic acid, known for its pronounced antimicrobial activity [30].

In our study, women with normal vaginal microbiota (Nugent 0–3) with complaints of discharge and itching comprised the largest group. According to the results of molecular profiling, lactobacilli predominated in the vaginal discharge of these patients. These complaints are believed to be related to lactobacillosis and cytolytic vaginitis—conditions debated among researchers [36] and characterized by lactobacilli excess with or without associated cytolysis. Studies demonstrate an association between the overgrowth of *L. crispatus* and increased acid production [37]. A large number of lactobacilli are responsible for maintaining an acidic microenvironment, sometimes accompanied by epithelial cytolysis. Vaginal smear microscopy reveals an increased number of lactobacilli and intracellular components. Clinical manifestation is often similar to that of vulvovaginal candidiasis [38]. To avoid misdiagnosis, accurate laboratory testing, based not solely on clinical complaints and microscopic findings, is certainly required.

A decrease in vaginal Lactobacilli is not always accompanied by an overgrowth of anaerobic microorganisms. Our test, in addition to BV, allows the identification of such conditions as vulvovaginal candidiasis, aerobic vaginitis, or intermediate vaginal microbiota. BV is characterized by a heterogeneous anaerobic composition and bacterial biofilms, contributing to BV persistence. At the same time, different microorganisms exert an effect on the clinical presentation of BV and treatment modalities [39].

The vaginal microbiota of reproductively aged women is constantly changing during menstrual cycle and throughout life. However, the predominance of lactobacilli in vaginal biotopes represents a hallmark feature of female health. Lactobacilli-deficient vaginal microbiota is associated with increased risks of BV, higher susceptibility to STDs, and possible pregnancy complications. Despite the fact that smear microscopy based on the Nugent scoring method is still considered the “method of choice” for BV diagnosis, the application of molecular tests allows more accurate determination of microorganisms number and diversity. Viral and STD pathogens detection is an important advantage for multiplex nucleic acid amplification tests, necessitating its wider introduction into routine laboratory practice.

## 4. Materials and Methods

### 4.1. General Study Design

The study was approved by the ethics committee of “The Research Institute of Obstetrics, Gynecology, and Reproductive Medicine named after D. O. Ott” (protocol code 108, dated 4 April 2021) and performed at the Department of Medical Microbiology. In total, 331 women aged 18–48 y. o. were recruited. 

The exclusion criteria were the following: pregnancy, lactation, menopause in women under 48 y. o., systemic and/or local antibacterial and probiotic therapy in the past 4 weeks, acute or chronic pelvic inflammatory diseases, severe somatic pathology, and malignancy at any localization. 

### 4.2. General Characteristics of Patients Included

A total of 331 reproductively aged, non-pregnant women were examined. A total of 301 women complaining of vaginal discharge, vaginal discomfort, and burning comprised the main group. Eighty-five percent of the women claimed vaginal discharge. The presence of an unpleasant odor was the second most common complaint–33% of cases. Thirty percent of women experienced vaginal itching and burning. The Nugent scoring method was used for microscopic examination of vaginal discharge. Under microscopic evaluation of Gram-stained smears, the following bacteria morphotypes were determined: large Gram-positive rods (*Lactobacillus*), small Gram-negative or Gram-variable curved rods (*Gardnerella* and *Bacteroides*), and Gram-negative or Gram-variable curved rods (*Mobiluncus*). Each of these parameters received a score based on the number of bacteria counted with subsequent total score calculation. Based on the scoring, samples were identified as: BV-negative (0–3), intermediate (4–6), and BV-positive (7–10).

According to the Nugent classification of vaginal samples, 232 (70%) samples were referred to as negative, 33 (10%)–intermediate, and 66 (20%) samples corresponded to BV.

A total of 331 samples of vaginal smears were examined by Femoflor test. Two of them were considered invalid (less than 4 lg–10.000 DNA copies). As a result, 329 samples were analyzed: 230 samples—negative for BV, 33—intermediate, 66—BV.

Thus, the main group was divided into three subgroups according to the Nugent criteria: BV-positive (7–10 points)—66 women, intermediate (4–6 points)—33 women, and BV-negative (0–3 points)—200 women. The control group consisted of 30 female healthcare providers with no complaints undergoing annual gynecological check-ups (PAP smear, microscopic examination of vaginal discharge, PCR for high-risk HPV). According to the Nugent system, all controls were scored 0–1 under microscopic examination.

Patients of both groups were comparable by age (31.09 ± 7.37 and 31.43 ± 6.93 years, respectively).

### 4.3. Samples Testing

For the purpose of BV diagnosis, microscopic examination of vaginal smears in accordance with the Nugent criteria was carried out. 

For the purpose of DNA extraction, DNA-sorb-AM kits were used (“NextBio” LLC, Moscow, Russia); DTPRIME amplifiers (“DNA-Technology” LLC, Moscow, Russia) were used for the reaction’s initiation.

A quantitative assessment of the total vaginal bacterial mass was carried out using the multiplex REAL-TIME PCR Detection Kit—Femoflor. The implemented PCR method is based on the amplification of a target DNA sequence using one biological sample and is expressed in genomic equivalent (GE). GE, in turn, is defined as the amount of DNA necessary to be present in the vaginal biotope to guarantee that all genes will be present and is expressed in decimal logarithms. A quantitative analysis of total bacterial count and genius/species-specific DNA of *Lactobacillus* spp., (*L. crispatus*, *L. iners*, *L. gasseri, L. jensenii*), *Bifidobacterium* spp., facultative anaerobic microorganisms (*Staphylococcus* spp., *Streptococcus* spp., Enterobacterales, *Enterococcus* spp., *Haemophilus* spp.), obligate anaerobic microorganisms (*Gardnerella vaginalis*, *Mobiluncus* spp., *Atopobium vaginae* (*Fannyhessea vaginae*), *Anaerococcus* spp., *Bacteroides* spp./*Porphyromonas* spp./*Prevotella* spp.; *Sneathia* spp./*Leptotrihia* spp./*Fusobacterium* spp.; *Megasphaera* spp./*Veilonella* spp./*Dialister* spp.; *Clostridium* spp./*Lachnobacterium* spp.; *Peptostreptococcus* spp.), as well as mollicutes *(Candida* spp., *Candida albicans*, *Ureaplasma urealyticum*, *Ureaplasma parvum*, *Mycoplasma hominis)*; STD pathogens (*Mycoplasma genitalium*, *Chlamydia trachomatis*, *Neisseria gonorrhoeae*, *Trichomonas vaginalis*); viruses (HSV1, HSV2, CMV, HPV) was automatically obtained. The reliability of the result was based on the control of sampling; in addition, a number of lactobacilli/total bacterial count (TBC) ratio was assessed in order to evaluate normal vaginal microbiota or dysbiosis and to compare the number of species to TBM in order to determine their etiological significance in aerobic, anaerobic or mixed dysbiosis.

### 4.4. Statistical Analysis

All statistical analyses were performed with the R 4.3.0 software environment. Nominal data (bacteria or virus positivity/negativity) were evaluated with Fisher’s exact test; if necessary Monte Carlo approximation was used, followed by intergroup comparison. Analysis of quantitative data (decimal logarithm of concentration) was carried out using a post-hoc test for Kruskal-Wallis. Quantitative data were presented as median and quartiles (additionally—mean and standard deviation). The median and quartiles were chosen as descriptive statistics due to the small number of samples within the groups. Confidence intervals were calculated using the Agresti-Coull method. Multiple comparisons were corrected with the Benjamini and Hochberg method. For all tests, a *p*-value of <0.05 was considered statistically significant.

In order to evaluate the prognostic ability of potential bacterial markers determined by the new Femoflor test to distinguish samples as normal microbiota or BV, ROC analysis was performed. The results between the Femoflor test and the Nugent method were compared using a two-way frequency table.

## Figures and Tables

**Figure 1 ijms-24-15880-f001:**
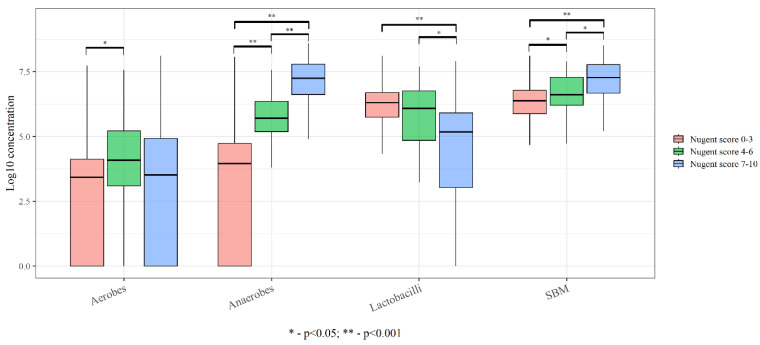
Comparative analysis of bacterial load in different types of vaginal microbiota.

**Figure 2 ijms-24-15880-f002:**
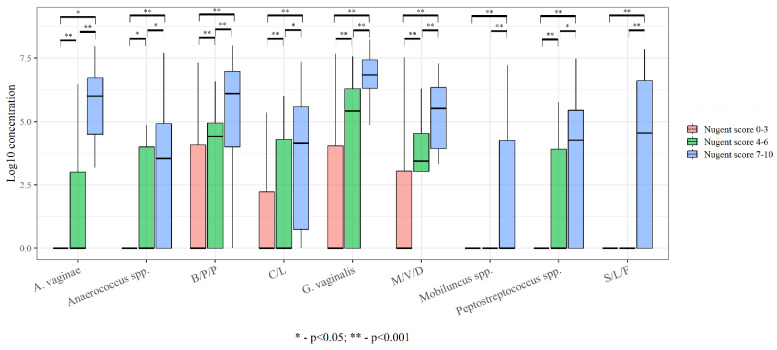
The detection rate and the number of obligate anaerobes in women with different types of vaginal microbiota. B/P/P-*Bacteroides* spp./*Porphyromonas* spp./*Prevotella* spp.; S/L/F-*Sneathia* spp./*Leptotrihia* spp./*Fusobacterium* spp.; M/V/D-*Megasphaera* spp./*Veilonella* spp./*Dialister* spp.; C/L-*Clostridium* spp./*Lachnobacterium* spp.

**Figure 3 ijms-24-15880-f003:**
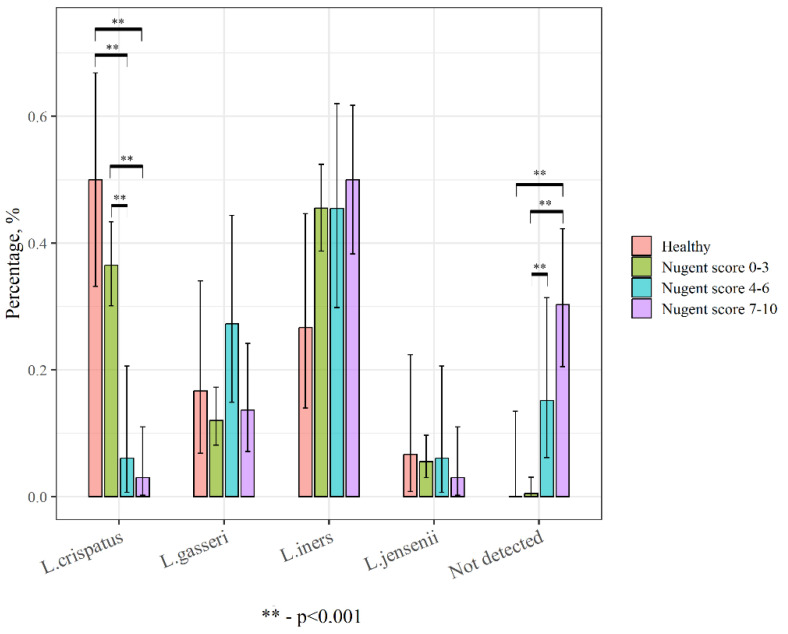
Detection rate of lactobacilli species in different types of vaginal microbiota.

**Figure 4 ijms-24-15880-f004:**
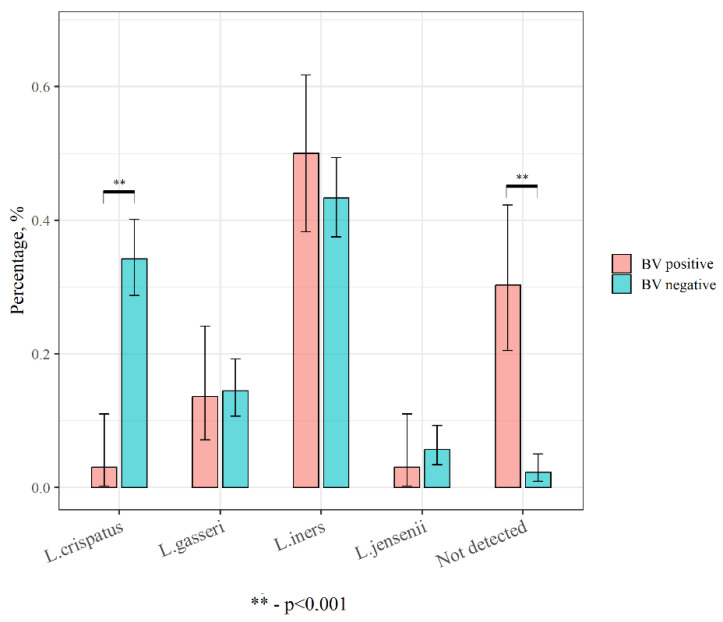
Dominant members of vaginal lactobacilli species in women with and without BV.

**Figure 5 ijms-24-15880-f005:**
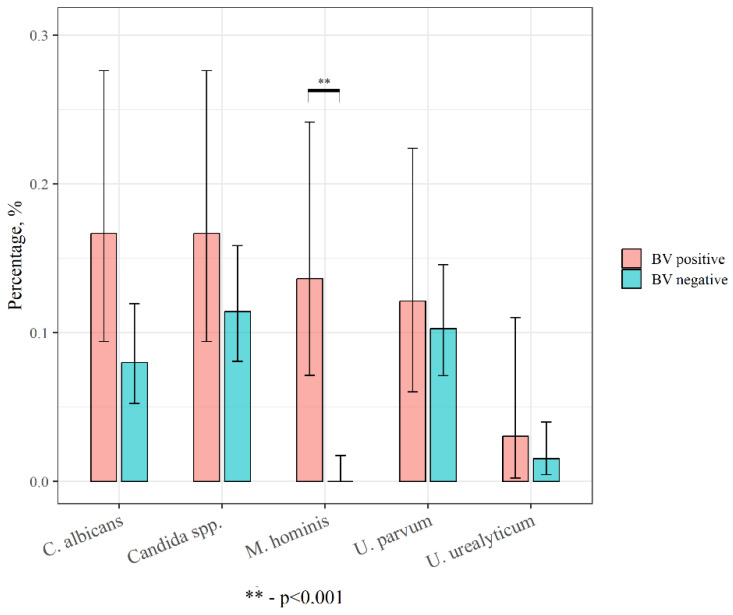
Detection rate of *Candida* spp., *Candida albicans,* and mollicutes in women with and without BV.

**Figure 6 ijms-24-15880-f006:**
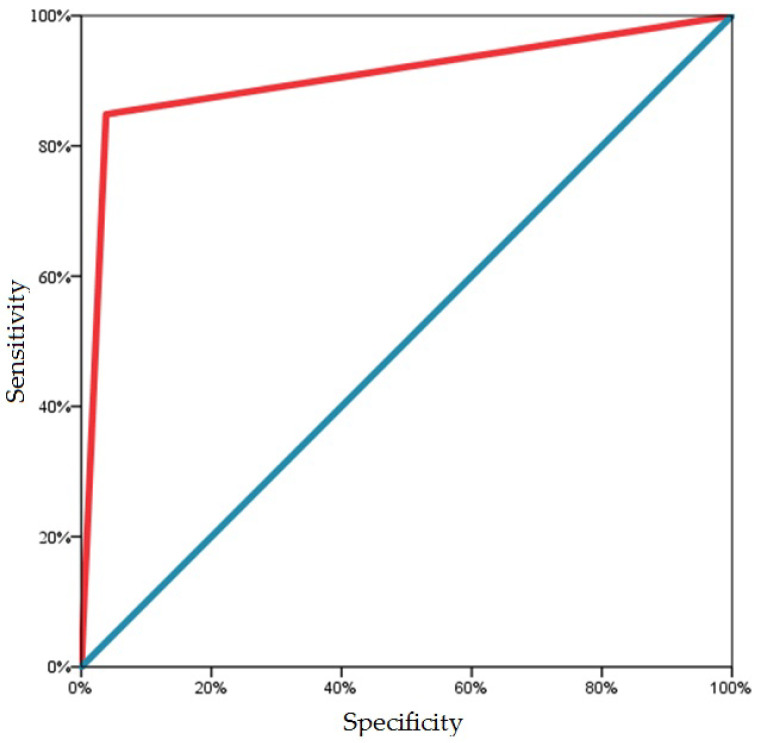
ROC analysis for Femoflor test in BV prediction.

**Table 1 ijms-24-15880-t001:** Microorganisms’ detection rate in the vaginal biotope of women with different types of microbiota according to Nugent’s criteria.

Microorganism	Study Group				Sig.
	Main Group			Control Group	
	Nugent Score 7–10 BV	Nugent Score 4–6 Intermediate	Nugent Score 0–3 Negative	Healthy	
**Normal microbiota**					
*Lactobacillus* spp.	46/66 (69.7%) ***	28/33 (84.8%)	198/200 (99%)	30/30 (100%)	3 vs. 4; *p* > 0.999 2 vs. 4; *p* = 0.081 1 vs. 4; *p* < 0.001
*L. crispatus*	15/66 (22.7%) ***	10/33 (30.3%)	121/200 (60.5%)	20/30 (66.7%)	3 vs. 4; *p* = 0.554 2 vs. 4; *p* = 0.008 1 vs. 4; *p* < 0.001
*L. iners*	37/66 (56.1%)	16/33 (48.5%)	129/200 (64.5%)	14/30 (46.7%)	3 vs. 4; *p* = 0.254 2 vs. 4; *p* > 0.999 1 vs. 4; *p* = 0.631
*L. gasseri*	11/66 (16.7%)	9/33 (27.3%)	51/200 (25.5%)	8/30 (26.7%)	3 vs. 4; *p* > 0.999 2 vs. 4; *p* > 0.999 1 vs. 4; *p* = 0.576
*L. jensenii*	12/66 (18.2%) *	8/33 (24.2%)	99/200 (49.5%)	14/30 (46.7%)	3 vs. 4; *p* = 0.846 2 vs. 4; *p* = 0.107 1 vs. 4; *p* = 0.016
*Bifidobacterium* spp.	24/66 (36.4%)	4/33 (12.1%)	23/200 (11.5%)	4/30 (13.3%)	3 vs. 4; *p* > 0.999 2 vs. 4; *p* > 0.999 1 vs. 4; *p* = 0.057
**Facultative anaerobic microorganisms (aerobes)**					
*Staphylococcus* spp.	9/66 (13.6%)	6/33 (18.2%)	54/200 (27%)	6/30 (20%)	3 vs. 4; *p* = 0.676 2 vs. 4; *p* > 0.999 1 vs. 4; *p* = 0.676
*Streptococcus* spp.	30/66 (45.5%)	21/33 (63.6%)	119/200 (59.5%)	11/30 (36.7%)	3 vs. 4; *p* = 0.125 2 vs. 4; *p* = 0.125 1 vs. 4; *p* = 0.608
*Enterobacteriales*	20/66 (30.3%)	12/33 (36.4%)	82/200 (41%)	10/30 (33.3%)	3 vs. 4; *p* = 0.977 2 vs. 4; *p* > 0.999 1 vs. 4; *p* = 0.977
*Enterococcus* spp.	5/66 (7.6%)	12/33 (36.4%) *	23/200 (11.5%)	2/30 (6.7%)	3 vs. 4; *p* = 0.655 2 vs. 4; *p* = 0.012 1 vs. 4; *p* > 0.999
*Haemophilus* spp.	23/66 (34.8%) **	4/33 (12.1%)	23/200 (11.5%)	1/30 (3.3%)	3 vs. 4; *p* = 0.428 2 vs. 4; *p* = 0.428 1 vs. 4; *p* = 0.002
**Obligate anaerobic microorganisms**					19
*Gardnerella vaginalis*	63/66 (95.5%) ***	23/33 (69.7%)	70/200 (35%)	13/30 (43.3%)	3 vs. 4; *p* = 0.417 2 vs. 4; *p* = 0.053 1 vs. 4; *p* < 0.001
*Mobiluncus* spp.	23/66 (34.8%) *	1/33 (3%)	13/200 (6.5%)	3/30 (10%)	3 vs. 4; *p* = 0.535 2 vs. 4; *p* = 0.51 1 vs. 4; *p* = 0.026
*Atopobium vaginae* (*Fannyhessea vaginae*)	52/66 (78.8%) ***	9/33 (27.3%)	22/200 (11%)	5/30 (16.7%)	3 vs. 4; *p* = 0.373 2 vs. 4; *p* = 0.373 1 vs. 4; *p* < 0.001
*Anaerococcus* spp.	44/66 (66.7%) **	16/33 (48.5%)	47/200 (23.5%)	9/30 (30%)	3 vs. 4; *p* = 0.494 2 vs. 4; *p* = 0.238 1 vs. 4; *p* = 0.003
*Bacteroides* spp./*Porphyromonas* spp./*Prevotella* spp.	51/66 (77.3%) **	21/33 (63.6%)	66/200 (33%)	12/30 (40%)	3 vs. 4; *p* = 0.535 2 vs. 4; *p* = 0.119 1 vs. 4; *p* = 0.003
*Sneathia* spp./*Leptotrihia* spp./*Fusobacterium* spp.	39/66 (59.1%) ***	6/33 (18.2%)	18/200 (9%)	3/30 (10%)	3 vs. 4; *p* = 0.743 2 vs. 4; *p* = 0.575 1 vs. 4; *p* < 0.001
*Megasphaera* spp./*Veilonella* spp./*Dialister* spp.	52/66 (78.8%) ***	25/33 (75.8%) ***	53/200 (26.5%)	7/30 (23.3%)	3 vs. 4; *p* = 0.826 2 vs. 4; *p* < 0.001 1 vs. 4; *p* < 0.001
*Clostridium* spp./*Lachnobacterium* spp./*bvab 23*	49/66 (74.2%) ***	14/33 (42.4%) *	53/200 (26.5%)	5/30 (16.7%)	3 vs. 4; *p* = 0.367 2 vs. 4; *p* = 0.047 1 vs. 4; *p* < 0.001
*Peptostreptococcus* spp.	47/66 (71.2%) ***	15/33 (45.5%)	32/200 (16%)	7/30 (23.3%)	3 vs. 4; *p* = 0.305 2 vs. 4; *p* = 0.133 1 vs. 4; *p* < 0.001
**Yeast**					
*Candida* spp.	11/66 (16.7%)	1/33 (3%)	29/200 (14.5%)	0/30 (0%)	3 vs. 4; *p* = 0.055 2 vs. 4; *p* > 0.999 1 vs. 4; *p* = 0.055
*Candida albicans*	11/66 (16.7%)	1/33 (3%)	20/200 (10%)	0/30 (0%)	3 vs. 4; *p* = 0.168 2 vs. 4; *p* > 0.999 1 vs. 4; *p* = 0.094
**Mollicutes**					
*Ureaplasma urealyticum*	2/66 (3%)	1/33 (3%)	3/200 (1.5%)	0/30 (0%)	3 vs. 4; *p* > 0.999 2 vs. 4; *p* > 0.999 1 vs. 4; *p* > 0.999
*Ureaplasma parvum*	8/66 (12.1%)	6/33 (18.2%)	17/200 (8.5%)	4/30 (13.3%)	3 vs. 4; *p* = 0.813 2 vs. 4; *p* = 0.882 1 vs. 4; *p* > 0.999
*Mycoplasma hominis*	9/66 (13.6%)	0/33 (0%)	0/200 (0%)	0/30 (0%)	3 vs. 4; *p* > 0.999 2 vs. 4; *p* > 0.999 1 vs. 4; *p* = 0.106
**Pathogenic microorganisms**					
*Mycoplasma genitalium*	1/66 (1.5%)	0/33 (0%)	0/200 (0%)	0/30 (0%)	3 vs. 4; *p* > 0.999 2 vs. 4; *p* > 0.999 1 vs. 4; *p* > 0.999
*Chlamydia trachomatis*	2/66 (3%)	0/33 (0%)	0/200 (0%)	0/30 (0%)	3 vs. 4; *p* > 0.999 2 vs. 4; *p* > 0.999 1 vs. 4; *p* > 0.999
*Neisseria gonorrhoeae*	1/66 (1.5%)	0/33 (0%)	0/200 (0%)	0/30 (0%)	3 vs. 4; *p* > 0.999 2 vs. 4; *p* > 0.999 1 vs. 4; *p* > 0.999
*Trichomonas vaginalis*	0/66 (0%)	0/33 (0%)	0/200 (0%)	0/30 (0%)	–
Herpes simplex virus I	0/66 (0%)	0/33 (0%)	0/200 (0%)	0/30 (0%)	–
Herpes simplex virus II	0/66 (0%)	1/33 (3%)	2/200 (1%)	0/30 (0%)	3 vs. 4; *p* > 0.999 2 vs. 4; *p* > 0.999 1 vs. 4; *p* > 0.999
Cytomegalovirus	0/66 (0%)	1/33 (3%)	3/200 (1.5%)	1/30 (3.3%)	3 vs. 4; *p* = 0.688 2 vs. 4; *p* > 0.999 1 vs. 4; *p* = 0.688
Human papillomavirus	14/66 (21.2%)	7/33 (21.2%)	53/200 (26.5%)	4/30 (13.3%)	3 vs. 4; *p* = 0.772 2 vs. 4; *p* = 0.772 1 vs. 4; *p* = 0.772

Two-tailed Fisher’s exact test, Benjamini and Hochberg method. *—*p* < 0.05; **—*p* < 0.01; ***—*p* < 0.001 when compared to healthy patients. 1 “Nugent score 7–10”; 2 “Nugent score 4–6”; 3 “Nugent score 0–3”; 4 “Healthy”.

**Table 2 ijms-24-15880-t002:** ROC analysis. Area under the curve (AUC).

AUC	*p*-Value	95% Confidence Interval	
		Lower Bound	Upper Bound
0.905	<0.001	0.853	0.958

**Table 3 ijms-24-15880-t003:** The sensitivity and specificity of Femoflor test in BV diagnosis.

Nugent’s Criteria		BV (Femoflor)		Total	Sensitivity	Specificity
		No	Yes			
BV	No	253	10	263	84.8%	96.2%
	Yes	10	56	66		
Total		263	66	329		

## Data Availability

The data that support the findings of this study are available on request from the corresponding author. The data are not publicly available due to privacy or ethical restrictions.

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
