# Peer review of "Vaginal Microbiota Molecular Profiling in Women with Bacterial Vaginosis: A Novel Diagnostic Tool"

_ijms, 2023, doi:10.3390/ijms242115880_

Round 1

Reviewer 1 Report

Comments and Suggestions for Authors

The paper is focused on the use of a new, Femflor test to detect BV in non-pregnant women. Generally, the paper is important in the field of BV diagnostics, it is well organised, with good statistics. However, some issues need further explanation.

1) Authors claim that patients were included on the basis of Nugent criteria, but these criteria are not described either in Introduction, nor in materials and methods.

2) Introduction is missing important information (Amsel, Nugent criteria - what they are), CSTII - what does this means to healthy vaginal microbiota?

3) The graphs in the present form are absolutely unacceptable - it is difficult to understand where are the values statistically significant, and the use of colors needs correction.

4)  Basic information on patients are missing - such as mean age, use of contraceptives, previous history of BV, etc. The patients cohort should be described  in more details.

5) who and on what basis (Nugent) was enrolling the patients? why only 30 controls were used? How the controls were chosen, on which basis - this is highly unclear and needs further explanation.

6) the specifity of Femflor should be compared by NGS. Description of the controls already included by the manufacturer belongs to the methods, rather to the scientific soundness of data.

7) which types of HPV were detected?

Comments on the Quality of English Language

The quality of English is good, the paper reads well but  multiply small mistakes must be corrected, such as unnecessary commas, sometimes clumsy pharases, missing words, etc.

For example - line 278 - "A total of 331 reproductively aged" - at reproductive age or?

Author Response

Thank you very much for taking the time to review this manuscript. Please find the detailed responses below and the corresponding revisions in the re-submitted files

Reviewer 2 Report

Comments and Suggestions for Authors

In this work, the authors developed a new molecular tool, based on RT-PCR, to detect and diagnose bacterial vaginosis (BV). The details of the new methods are well described and compared with currently available standard methods and the authors’ conclusions are supported by the reported data. Thus, we recommend publication, after addressing the few minor points below:

-          Line 57: please briefly describe what the Nugent scoring method consists of, how does it work?

-          Line 108: how was the total bacterial mass calculated ? please also add units on the relative graph y-axis (log 10 of what ?)

-          If I well understood, the term “negative” is used to describe different classes of situations (e.g. BV-negative; lactobacilli-negative) this generates a bit of confusion. I suggest to indicate these relative classes more explicitly (e.g. BV-negative; lactobacilli-negative). Also, for consistency, I would indicate the class “BV” as “BV-positive”.

-          Same comment in Figure 3 and 4: BV yes and BV no, could be renamed as BV-positive and BV negative, respectively.

Figures can be improved and made more legible, few suggestions:

-          Line 110 (Figure 1): legend for classes 1 and 2 can be added directly in the figure legend on the right side of the plot (e.g: BV (1); Negative (2) …). Even better, if the statistical significance is reported in the plot in the classical way (with a line above two bars being compared and the *) there is no need for this additional classification which creates a bit of confusion. Same comment for all subsequent Figures and plots.

-          Why the bars are partially superimposed ? It would be easier to read the graph if they were distinct and separated

-          Additional pie charts, one for each class (BV yes; BV no etc) in Figures 3, 4 and 5 may help to better understand the situation

Comments on the Quality of English Language

English language minor revision:

-          line 58: “diagnosing” should be “diagnosis”

-          line 64-5: “THEY determine ….and THEY are optimal…” please add “they”

-          line 182: “analysis” should be “analyses” or “to THE molecular analysis…”

-          line 224: “ A PREVIOUS Australian study, indicating exclusively M. hominis association…” please add “ a previous” and remove “the” and “on”

-          line229-30: “our results can be POSSIBLY explained” please change syntaxis accordingly

Author Response

Thank you very much for taking the time to review this manuscript. Please find the detailed responses below and the corresponding revisions in the re-submitted file
